# Computational epitope map of SARS-CoV-2 spike protein

**Mateusz Sikora**[1,2☯], **Sören von Bülow**[1☯], **Florian E. C. Blanc**[1☯], **Michael Gecht**[1☯], **Roberto Covino**[1,3☯], **Gerhard Hummer**[1,4]*

**1** Department of Theoretical Biophysics, Max Planck Institute of Biophysics, Frankfurt am Main, Germany, **2** Faculty of Physics, University of Vienna, Vienna, Austria, **3** Frankfurt Institute for Advanced Studies, Frankfurt am Main, Germany, **4** Institute of Biophysics, Goethe University Frankfurt, Frankfurt am Main, Germany

☯ These authors contributed equally to this work.
* gerhard.hummer@biophys.mpg.de

## Abstract

The primary immunological target of COVID-19 vaccines is the SARS-CoV-2 spike (S) protein. S is exposed on the viral surface and mediates viral entry into the host cell. To identify possible antibody binding sites, we performed multi-microsecond molecular dynamics simulations of a 4.1 million atom system containing a patch of viral membrane with four full-length, fully glycosylated and palmitoylated S proteins. By mapping steric accessibility, structural rigidity, sequence conservation, and generic antibody binding signatures, we recover known epitopes on S and reveal promising epitope candidates for structure-based vaccine design. We find that the extensive and inherently flexible glycan coat shields a surface area larger than expected from static structures, highlighting the importance of structural dynamics. The protective glycan shield and the high flexibility of its hinges give the stalk overall low epitope scores. Our computational epitope-mapping procedure is general and should thus prove useful for other viral envelope proteins whose structures have been characterized.

## Author summary

The SARS-CoV-2 virus has caused a global health crisis. The spike protein exposed at its surface is key for infection and the primary antibody target. However, spike is covered by highly mobile glycan molecules that could impair antibody binding. To identify accessible epitopes, we performed molecular dynamics simulations of an atomistic model of glycosylated spike embedded in a membrane. By combining extensive simulations with bioinformatics analyses, we recovered known antibody binding sites and identified several epitope candidates as targets for further vaccine development.

## Introduction

The ongoing COVID-19 pandemic, caused by the SARS-CoV-2 coronavirus, has emerged as the most challenging global health crisis within a century [1]. Vaccination is the most

**Data Availability Statement:** The structure and GROMACS topology files used to simulate the system are available at: https://doi.org/10.5281/zenodo.3906317 All other relevant data are within

the manuscript and its Supporting information files.

**Funding:** This work was supported by the Max Planck Society (https://www.mpg.de) (GH), the Austrian Science Fund FWF Schrödinger Fellowship J4332-B28 (https://www.fwf.ac.at) (MS), the Human Frontier Science Program RGP0026/2017 (https://www.hfsp.org) (GH), the Landes-Offensive zur Entwicklung Wissenschaftlich-Ökonomischer Exzellenz LOEWE of the State of Hesse (https://wissenschaft.hessen.de/wissenschaft/landesprogramm-loewe): DynaMem (GH) and CMMS (RC and GH), the Frankfurt Institute for Advanced Studies (https://fias.institute): (RC), and the Leibniz Supercomputing Centre Munich (https://www.lrz.de): SUPERspike (GH). The funders had no role in study design, data collection and analysis, decision to publish, or preparation of the manuscript.

**Competing interests:** No authors have competing interests.

promising strategy to end the pandemic. As for other enveloped viruses [2], the primary vaccine target is the trimeric spike (S) protein on the envelope of SARS-CoV-2. S mediates viral entry into the target cell [3–7]. After binding to the human angiotensin-converting enzyme 2 (ACE2) receptor, the ectodomain of S undergoes a drastic transition from a prefusion to a postfusion conformation. This transition drives the fusion between viral and host membranes, which triggers internalization of SARS-CoV-2 via endocytic and possibly non-endocytic pathways [8, 9]. Locking the prefusion conformation of S or blocking its interaction with ACE2 would prevent cell entry and infection, a task achieved by a growing number of neutralizing antibodies [10–16].

Structure-based rational design promises improvements in vaccine efficacy [17] and could lead to therapeutic cocktails that minimize the risk of immune evasion by using epitopes on non-overlapping regions of S [18]. A detailed understanding of the exposed viral surface will, therefore, be instrumental [17].

Thanks to the extraordinary response of the global scientific community, we already have atomistic structures of S [6, 7, 19, 20] and detailed views of the viral envelope [21–24]. However, static structures do not capture conformational changes of S or the motion of the highly dynamic glycans covering it. Molecular dynamics (MD) simulations add a dynamic picture of S and its glycan shield [23, 25–27]. Intriguingly, several groups have shown experimentally that glycans not only shield the S protein but also play a role in the infection mechanism [26, 28, 29].

Here, we report on the 2.5 μs-long MD simulation of a full-length atomistic model of four S trimers in the prefusion conformation, amounting to 10 μs of S dynamics. The model includes the transmembrane domain (TMD) embedded in a complex lipid bilayer, along with realistic post-translational modification patterns, i.e., glycosylation of the ectodomain and palmitoylation of the TMD. Although independently developed, we have recently shown that our S protein model and its structural dynamics are in quantitative agreement with recent high-resolution electron cryo-tomography (cryoET) reconstructions [23]. Intact virions present their S proteins either individually, in small groups, or in large clusters, in a strikingly random distribution [23]. To quantify the effect of this heterogeneous distribution, we analyzed the accessibility of S epitopes both in isolation (i.e., with surrounding S proteins removed) and in the dense packing of our simulation model.

We identify epitope candidates on SARS-CoV-2 S by combining information on steric accessibility and structural flexibility with bioinformatic assessments of sequence conservation and epitope characteristics. We recover known epitopes in the ACE2 receptor-binding domain (RBD) and identify several epitope candidates on the spike surface that are exposed, structured, and conserved in sequence. In particular, target sites for antibodies emerge in the functionally important S2 domain harboring the fusion machinery.

## Results

### Model of full-length S

To search for possible epitopes, we constructed a detailed structural model of glycosylated full-length S. Whereas high-resolution structures of the S head are available [6, 7], the stalk and membrane anchor have so far not been resolved at the atomic level.

We built a model of the complete S by combining experimental structural data and bioinformatic predictions. Our full-length model of the S trimer consists of the large ectodomain (residues 1-1137) forming the head, two coiled coil (CC) domains, denoted CC1 (residues 1138-1158) and heptad repeat 2 (HR2, residues 1167-1204), forming the stalk, the $\alpha$-helical TMD (residues 1212-1237) with flanking amphipathic helices (AH, 1243-1255) and multiple

palmitoylated cysteines, and a short C-terminal domain (CTD, residues 1256-1273), see S1(A) Fig for domain definitions. We modeled the glycosylation pattern as recently revealed for over-expressed S [30] (see S1(B) Fig). Despite passage through an intact Golgi, expressed glycans closely resemble those of native SARS-CoV S [31].

As shown [23], the model fits high-resolution cryoET electron density data of S proteins on the surface of virions extracted from a culture of infected cells remarkably well. It also captures the stalk domain with its three flexible hinges between the S head and CC1, CC1 and HR2, and HR2 and the TMD. The tomographic maps also confirmed the extensive glycosylation of the model [23].

## Multi-microsecond atomistic MD simulations reveal dynamics of S and its glycan shield

We performed a 2.5 μs long atomistic MD simulation of a viral membrane patch with four flexible S proteins, embedded at a distance of about 15 nm [32, 33] (Fig 1). During the simulation, the four S proteins remained folded (S2(G)–S2(J) Fig) and stably anchored in the membrane with well-separated TMDs.

The S heads tilted dynamically and interacted with their neighbors (S1 Movie). High-resolution cryoET images [23] and a recent MD study [26] independently revealed significant head tilting associated with the flexing of the joints in the stalk, in strong support of our observations. Being highly mobile, the glycans cover most of the S surface (Fig 2A–2C).

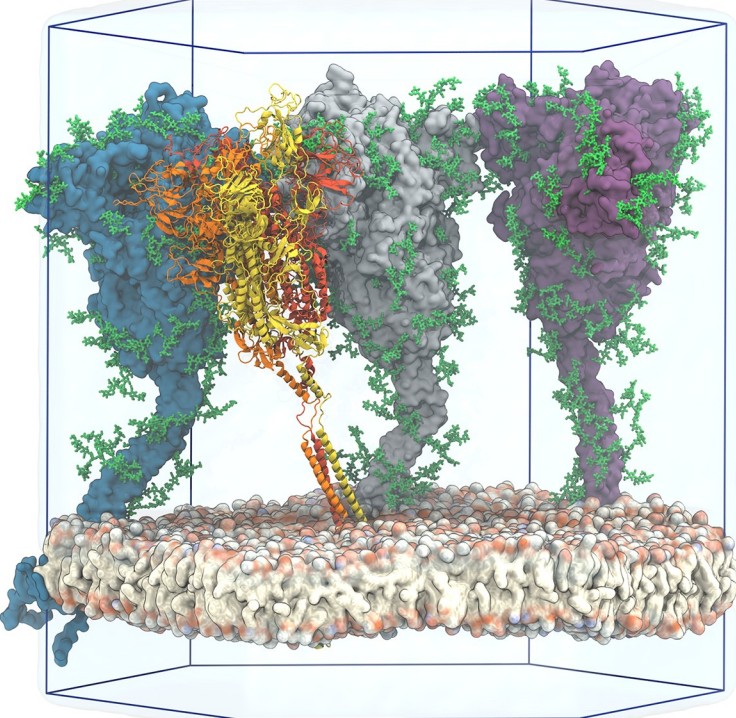

**Fig 1. View of the simulated atomistic model containing four glycosylated and membrane-anchored S proteins in a hexagonal simulation box.** Three proteins are shown in surface representation with glycans represented as green sticks. One protein is shown in cartoon representation, with the three chains colored individually and glycans omitted for clarity. Water is shown as a transparent blue surface and ions are omitted for clarity. Two simulation box edges are not drawn for better visibility.

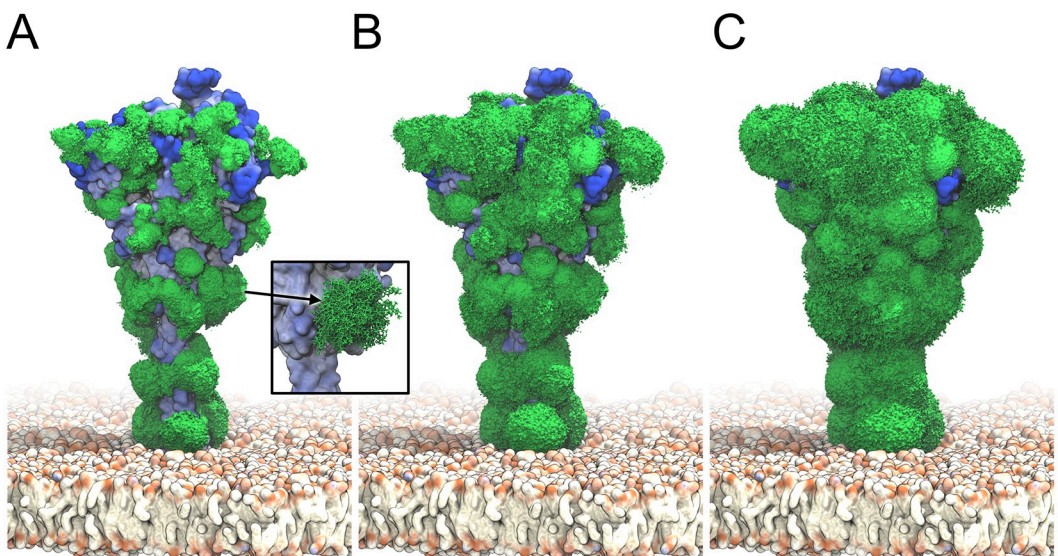

**Fig 2. S glycan dynamics from MD simulations.** Time-averaged glycan electron density isosurfaces are shown at high (*A*), medium (*B*), and low (*C*) contour levels, respectively. The blue-to-white protein surface indicates high-to-low accessibility in ray analysis. (Inset) Snapshots (sticks) of a biantennary, core-fucosylated and sialylated glycan at position 1098 along the MD trajectory.

## Antibody binding sites predicted from accessibility, rigidity, sequence conservation, and sequence signature

**Accessibility of the S ectodomain.** Antibody binding requires at least transient access to epitopes. The glycan shield covering the surface proteins of enveloped viruses can sterically hinder access to these binding sites, helping SARS-CoV-2 to evade a robust immune response [34]. We assessed the accessibility of S on the viral membrane and the surface coverage by glycans by (i) ray and (ii) antigen-binding fragment (Fab) docking analyses of the S configurations in our MD simulations. In the ray analysis, we illuminated the protein model by diffuse light; in the Fab docking analysis, we performed rigid-body Monte Carlo simulations of S configurations taken from the MD simulations together with the SARS-CoV-2 antibody CR3022 Fab to quantify how easily a Fab antibody could access the surface of S. To account for protein and glycan mobility, we performed both analyses individually for $4 \times 250$ snapshots taken at 10 ns time intervals from the 2.5 µs MD simulation with four glycosylated S proteins.

The dynamic glycan shield effectively covers the S surface (Fig 3A and 3B). Even though glycans cover only a small fraction of the protein surface at any given moment (Fig 1), their high mobility leads to a strong steric shielding of S (Fig 2). A comparison of the ray (S3(A)–S3(C) Fig) and Fab docking results (S3(D)–S3(I) Fig) for glycosylated and unglycosylated S illustrates this effect. We consider ray and docking analyses to be complementary: The ray analysis provides an upper bound to the accessibility because the thin rays can penetrate more easily through the glycan shield than antibodies, whereas the rigid-body docking gives a lower bound because it does not take into account any induced fit from interactions between glycans and antibody. Importantly, the two methods are consistent in identifying regions of high and low accessibility (Fig 4A and 4B). Ray and docking analyses show that glycans cause a reduction in accessibility by about 34% and 80%, respectively (Table B in S1 Text). The most marked effect occurs in the HR2 coiled coil close to the membrane. Without glycosylation, HR2 is fully accessible; with glycosylation, HR2 becomes inaccessible to Fab docking. Whereas small

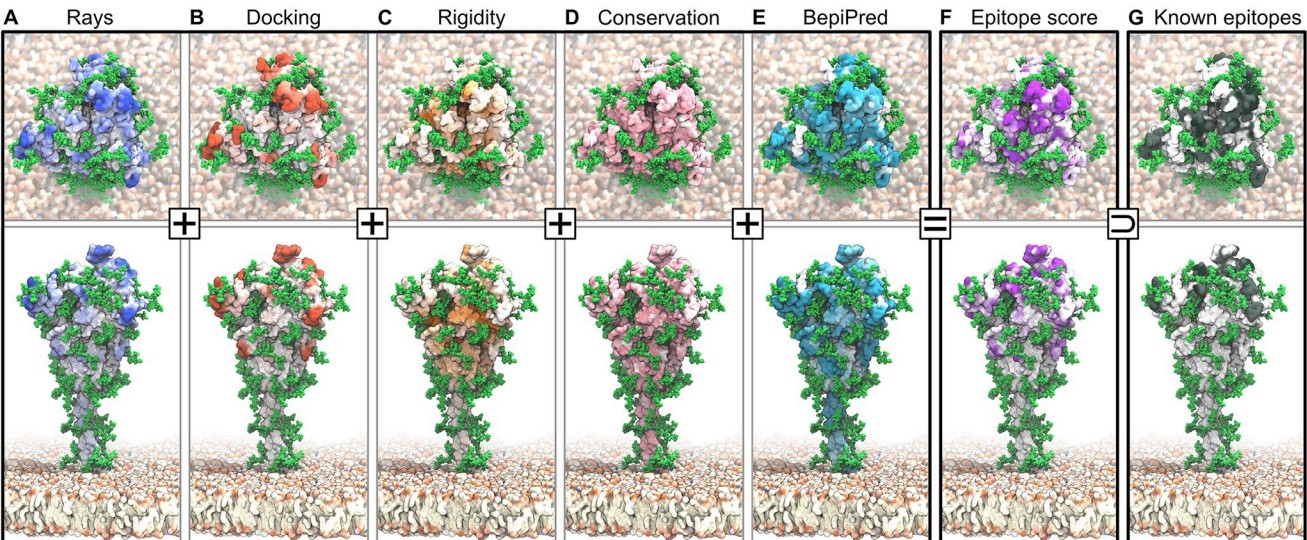

**Fig 3. Epitopes identified from MD simulations and bioinformatics analyses.** Accessibility scores from (*A*) ray analysis and (*B*) Fab rigid-body docking are combined with (*C*) rigidity scores, all averaged over 4 × 2.5 μs of S protein MD simulations. Also included are (*D*) a sequence conservation score [35], and (*E*) BepiPred-2.0 epitope sequence-signature prediction. (*F*) Combined epitope score. (*G*) Binding sites of known neutralizing antibodies. Higher color intensity in *A-F* indicates a higher score and higher color intensity in *G* indicates sites binding to multiple different antibodies.

molecules may interact with the HR2 protein stalk, antibodies are blocked from surface access, in agreement with recent independent simulations [26].

On SARS-CoV-2 virions, S proteins occasionally form dense clusters, which may enhance the avidity of the interactions with human host cells [23]. To quantify the effect of crowding, we compared the epitope accessibility of S from the ray and docking analyses in the dense simulation system (S3(C) and S3(I) Fig) and in isolation (S3(B) and S3(H) Fig), i.e., with the other proteins removed from the MD system. Overall, protein crowding reduced the accessibility of S by another ∼ 5% in the ray analysis and ∼ 6% in the Fab docking analysis, resulting in a combined accessibility reduction by glycans and crowded proteins of ∼ 39% and ∼ 86%, respectively.

**Rigidity of S.** Structured epitopes are expected to bind strongly and specifically to antibodies. By contrast, mobile regions tend to become structured in the bound state, entailing a loss in entropy and may not retain their structure when presented in a vaccine construct. With the aim of eliciting a robust immune response, we chose to include rigidity in our epitope score. Here, we focus on motions of domains on the scale of about 1 nm. We analyzed large-scale conformational dynamics associated with the flexible hinges in the stalk and membrane anchor in another paper [23]. We determined the root-mean-square fluctuations (RMSF) by superimposing protein structures and converting the RMSF into a rigidity score, as described in Methods.

The surface of S presents both dynamic and rigid regions (Fig 4C). Interestingly, the RBD and its surroundings are comparably flexible, consistent with the experimental finding of large differences in the structure of the three peptide chains in open and closed states [7]. By contrast, the protein surface of the S2 domain covering the fusion machinery is relatively rigid (Fig 4C), possibly to safeguard this functionally critical domain in the metastable prefusion conformation.

**Sequence conservation.** Targeting epitopes whose sequences are highly conserved will ensure efficacy across strains and prevent the virus from escaping immune pressure through

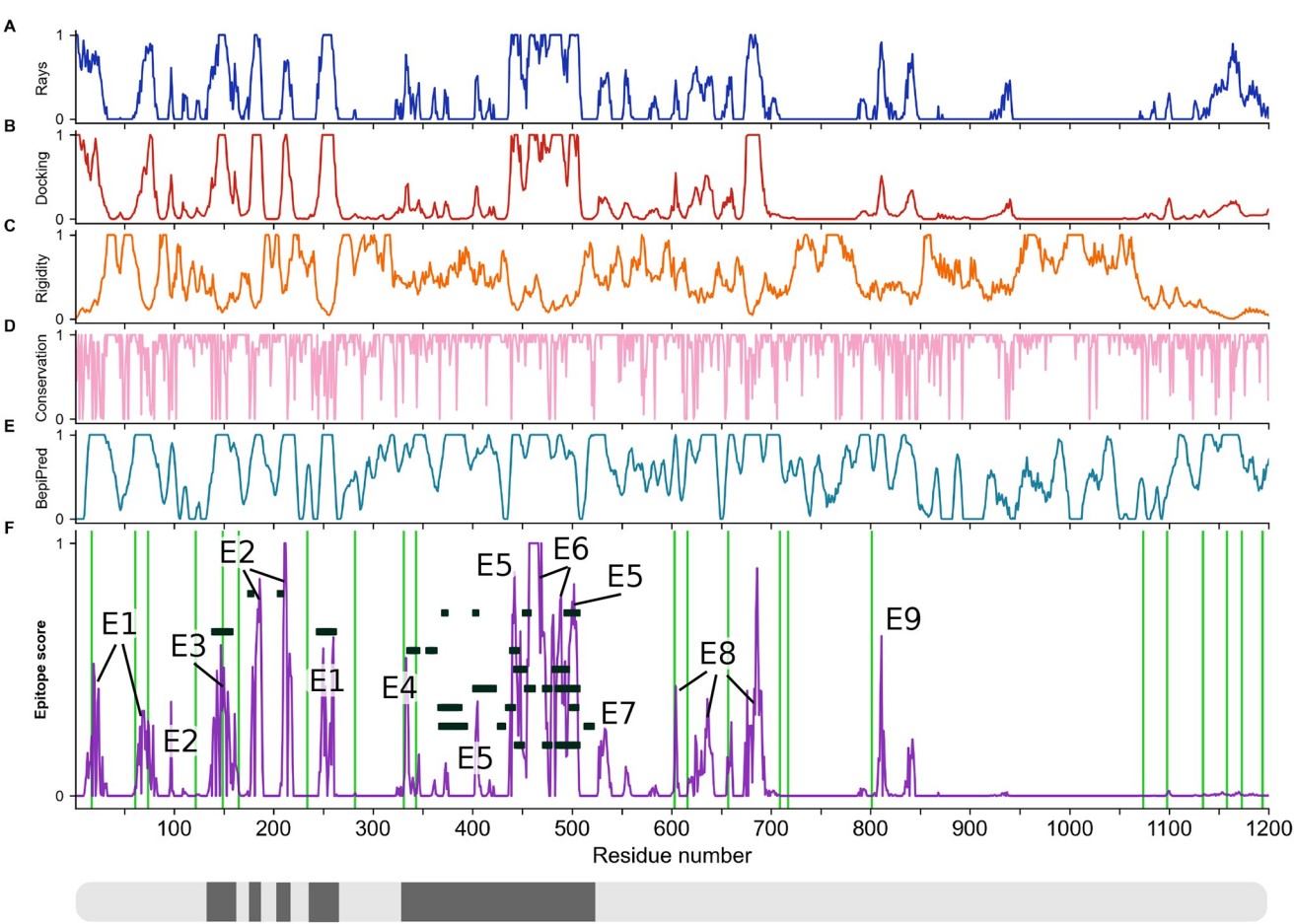

**Fig 4. Epitope scores of the S ectodomain.** Panels (A-F) and colors as in Fig 3. All values are filtered and normalized (see Methods). Labels E1–E9 in (F) highlight candidate epitopes. Green lines indicate glycosylation sites. Black rectangles show known antibody binding sites, also indicated in black along the S sequence in the bottom box.

mutations with minimal fitness penalty. We estimated the sequence conservation from the naturally occurring variations at each amino acid position in the sequences collected and curated by the GISAID initiative (https://www.gisaid.org/). The analysis of 30,426 amino acid sequences revealed that S is overall highly conserved, with no mutation recorded for 52% of the amino acid positions. As conservation score, we mapped the entropy at each position to the interval between zero and one (see Methods). Even surface regions are mostly well conserved in sequence (Fig 4D).

**Sequence-based immunogenicity predictor.** Conserved, rigid, and accessible regions present good candidates for binding of protein partners in general. To complement this information, we assessed the immunogenic potential based on sequence signatures targeted by antibodies. The epitope-like motifs in the S sequence identified by using the BepiPred 2.0 server [36] lie scattered across the S ectodomain and include known epitopes (Figs 3E and 4E), but also contain buried regions inaccessible to antibodies.

**Consensus epitope score.** We combined our accessibility, rigidity, conservation, and immunogenicity scores into a single consensus epitope score (Figs 3F and 4F). By taking the product of all individual scores, we ensured that epitope candidates have high scores in all features. This stringent requirement eliminates many candidate sites, mostly because accessibility

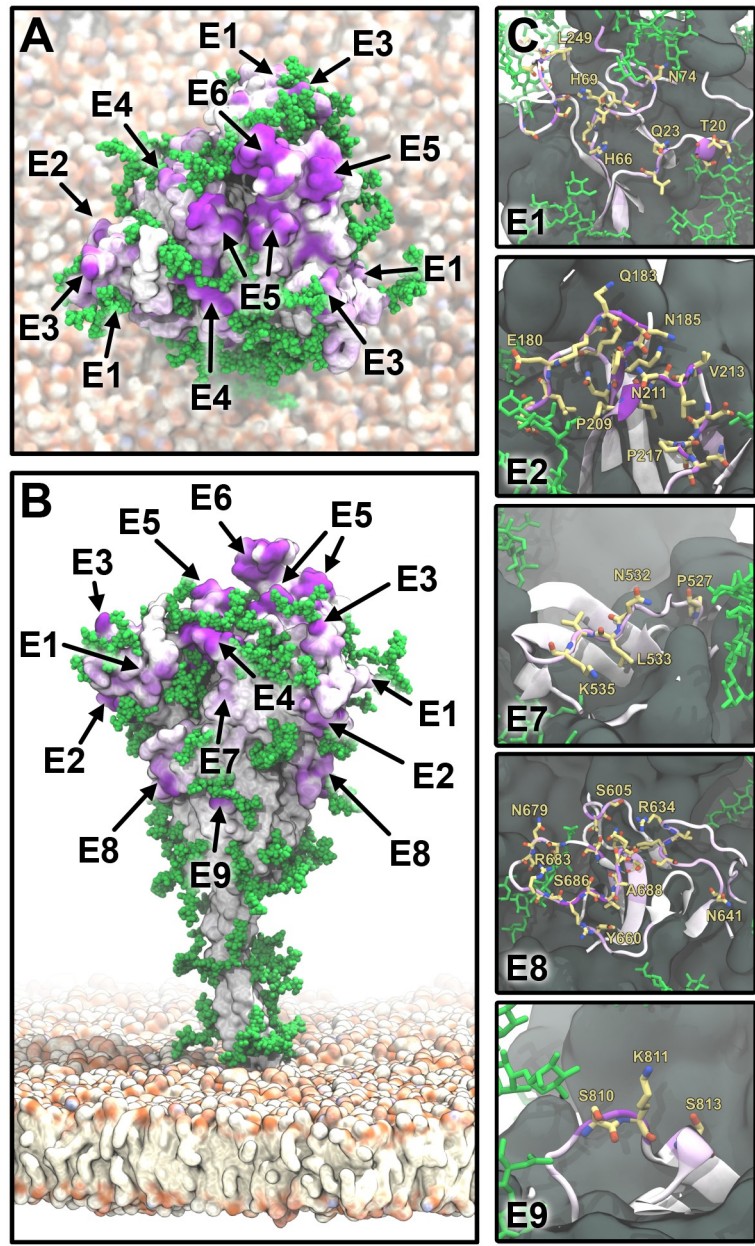

**Fig 5. S epitope candidates.** (*A*) Top view of S represented as in Fig 3F. Epitope candidates are labeled according to Table 1. (*B*) Side view with coloring and labels as in A. (*C*) Zoom-ins on epitope candidates (E1, E2, E7–E9) in a cartoon representation and colored as in A. Residues with an epitope consensus score >0.2 are shown in yellow licorice representation.

scores (Fig 4A and 4B) and the rigidity score (Fig 4C) show opposite trends, in line with the extensive occurrence of flexible loops on the S surface.

Using our consensus score, we identified nine epitope candidates (E1–E9; Fig 5 and Table 1). Epitope candidates E3–E6 recover known epitopes (Figs 3F, 3G and 4F), in some cases achieving residue-level accuracy (S4 Fig); in addition, we identify epitope candidates E1, E2, and E7–E9. All epitope candidates reside in the structured head of S. By contrast, low accessibility and high flexibility in the hinges [23] give the stalk low overall epitope scores.

**Table 1. Epitope candidates shown in Figs 3–5.**

| Epitope | Residues |
|---------|----------|
| E1 | 15-28, 63-79, 247-260 |
| E2 | 97, 178-189, 207-219 |
| E3 | 137-164 |
| E4 | 332-346 |
| E5 | 403-406, 438, 440-451, 495-506 |
| E6 | 452-476, 479-482, 484-494 |
| E7 | 527-537 |
| E8 | 603-605, 633-642, 656-661, 674-693 |
| E9 | 808-814 |

In our simulation, two of the RBD domains were sampled in the closed conformation and one in the open conformation. To assess the effect of open and closed states, we computed the accessibility and consensus scores for all eight protein chains with closed RBD conformation (S10 Fig). A comparison with Fig 4 highlights that the only sizable differences in accessibility occur in the RBD region.

Crowding of S causes a significant drop in the score of epitope candidates E1 and E2, whereas it has only little effect on the candidates E3–E4 and E7–E8, and no effect on the candidates E5–E6 and E9 (S5 Fig).

**Collective behavior of S.**   Despite the remarkably random distribution of S at the surface of the virions, densely populated patches are not uncommon (cf. Fig 5G in [23]). Multiple S will likely come in contact if simultaneously bound to a single ACE2 dimer or when cross-linked by antibodies. Taking advantage of our simulation setup, we analyzed interactions between S. During 2.5 μs we observed a number of contacts involving both glycans and protein surfaces, which resulted in partial jamming of three of the S in a characteristic triangular arrangement of S head domains and with the fourth S only weakly interacting with two of its neighbors (S9(A) Fig). In the jammed state glycans formed an extensive network of interactions, as could be seen in S9(B) Fig. To quantify relative roles of particular residues and glycan moieties we computed a contact map of inter-S interactions (S9(C) and S9(D) Fig). We found the majority of protein-mediated contacts to reside in the unstructured loops within NTD regions. Glycan-mediated contacts concentrate in the sequons located in the NTD and RBD areas and at the bottom of the S head. Despite their size, glycans located on the stalk were not involved in the inter-S contacts. Instead, they remained relatively shielded by the much more spacious head domains.

## Discussion

### Known antibody interactions validate the epitope identification procedure

A rapidly growing number of studies report on antibody binding to the S protein [10–16] and provide us with excellent reference data to validate our strategy for epitope identification. The focus in these studies has been on antibodies binding to the exposed RBD of S to achieve a high degree of neutralization by blocking binding to the ACE2 receptor. Yuan *et al.* structurally characterized the binding of SARS-CoV-neutralizing antibody CR3022 to the SARS-CoV-2 S protein ectodomain [10, 11, 37]. Their structure reveals an epitope distal to the ACE2 binding site that requires at least two of the S protomers to be in the open conformation to permit binding without steric clashes. Interestingly, while our simulations do not probe the doubly open configuration, the epitope reported by Yuan *et al.* [10] is still successfully identified with

a significant consensus score. Moreover, epitopes for other reported antibodies H014 [12], CB6 [13], P2B-2F6 [14], S309 [16], and 4A8 [38] also match regions of high consensus score. In particular, our candidate epitopes E5 and E6 overlap with the reported binding sites in the RBD for neutralizing antibodies [39–43]. We conclude that our epitope-identification methodology is robust.

## Dependence on detailed glycosylation pattern

Mass spectrometry on recombinant S indicated extensive glycosylation [30] with oligomannose, and sialylated and fucosylated hybrid and complex glycans. Despite recent cryoET images of intact viral particles confirming occupancy of the majority of sequons and revealing glycan branching [23], the extent and composition of glycans *in situ* remains poorly understood. Pre-Golgi budding of the virions [44], overloading cells with polysaccharide production and high density of viral glycans (also reported in HIV [45]) can all contribute to non-canonical and not fully matured glycans.

We addressed this uncertainty by repeating our docking accessibility analysis for different glycosylation patterns (S1(B) Fig). In addition to the "full" glycan pattern used in the simulations, we analyzed the accessibility in a resampled simulation with all sites occupied by the mannose-type glycans (Mannose-5, Man5). Remarkably, the reduced glycan shield impedes Fab accessibility almost as effectively ($\sim$75%) as the full shield ($\sim$80%), even if epitopes E7–E9 become somewhat more exposed with shorter glycans (S3(D)–S3(H) Fig). Interestingly, the largest and most processed complex glycans are found on the flexible stalk [30], suggesting that this region is critical for the viral cycle and must be shielded from the immune system. Overall, we conclude that even a light glycan coverage might hinder the antibody accessibility of the protein in a significant manner.

## Structural and dynamic characteristics of candidate epitopes

Epitopes E1–E3 are part of the N-terminal domain (NTD, residues 1-291), which is formed mostly of antiparallel $\beta$ sheets. All three epitopes include flexible loops and folded $\beta$ strands (S6(A) Fig). Interestingly, epitope E2 includes residue 207 (Table 1) and is in close proximity to residue 177, both of which have been reported by Schoof and co-workers [43] to be involved in binding an allosteric nanobody. We propose that E2 could represent the full binding site of this nanobody, which has not been mapped completely. Thus, our method may also be used to complement experimental characterizations of epitopes. Epitope E4 is located on a two-turn $\alpha$-helix flanked by a short twin $\alpha$-helix and lying on a five-strand antiparallel $\beta$-sheet. This arrangement provides the epitope with remarkable stability (S6(B) Fig). Epitopes E5 and E6 are located on the apical part of S in the RBD, and are composed mostly of flexible loops. E5 and E6 jointly span a contiguous surface in chain A, which is in the open conformation. By contrast, in the closed chains B and C, this surface is altered and E6 is buried (S6(C) Fig). The epitope E7 is part of a stable helix that connects neighboring $\beta$-sheets (S6(D) Fig). E8 comprises two quite long and flexible loops (residues 634-641 and 674-693), and two shorter and less flexible ones (S6(E) Fig). Finally, E9 is located on a short and flexible loop (S6(F) Fig).

## Glycans as epitopes

Even though glycans sterically hinder the accessibility of the surface of S, antibodies can in some instances tolerate the close proximity to glycans [34]. Moreover, glycans themselves can be part of epitopes in SARS-CoV-2 S [16, 26] and HIV-1 Env [46]. While this could open up possibilities for epitope binding, the natural variability of the glycan shield [30], along with its extensive structural dynamics demonstrated here, currently preclude a systematic search for

glycan-involving epitopes. Moreover, with human and viral proteins carrying chemically equivalent glycan coats, the risk of autoreaction is significant [46]. Therefore, we concentrated here on sterically accessible amino acid epitopes.

## Conclusions

We identified epitope candidates on the SARS-CoV-2 S protein surface by combining accurate atomistic modelling, multi-microsecond MD simulations, and a range of bioinformatics analysis methods. We concentrated on sites that are accessible to antibodies, unencumbered by the glycan shield, and fairly rigid. We also required these sites to be conserved in sequence and to display signatures expected to elicit an immune response. From all these features, we determined a combined consensus epitope score that predicts nine distinct epitope sites. Validating our methodology, we recovered five epitopes that overlap with experimentally characterized epitopes, including a "cryptic" site [10].

Highly dynamic glycans cover the S surface to a great extent and could produce immunogenic shielding by suppressing some interaction modes with antibodies. Even though the instantaneous surface coverage of the glycans is low, over time relatively few well placed glycans cover most of the protein surface. In particular, only three N-glycosylation sites per protein chain suffice to shield the stalk domain and block antibody binding to this functionally critical part of the protein. New and conflicting reports emerge on the glycan types on the S surface [30, 47], with glycan composition possibly varying from host to host. We considered both light and heavy glycan coverages in our analysis, which should encompass most of the glycan variability. We obtained an excellent correspondence in the glycan coverage in a direct comparison to high-resolution tomographic maps of S proteins on intact virions [23]. We found that already the light glycosylation sterically hinders the interaction between antibodies and S in a significant manner.

The different epitopes we predicted provide starting points to engineering stable immunogenic constructs that robustly elicit the production of antibodies. A fragment-based epitope presentation avoids the many challenges of working with full-length S, a multimeric and highly dynamic membrane protein, whose prefusion structure is likely metastable [48]. Epitopes E1, E2, E3, and E8 are particularly promising candidates. They are located on distinct S domains that could fold independently and present these epitopes in a native-like manner [49]. Mutational escape by SARS-CoV-2 can lead to loss of neutralization of specific antibodies [18]. The use of antibody cocktails targeting spatially distinct epitopes on S should suppress the development of resistance [18, 42, 50]. The approach we introduced in this paper is general and can be extended to predict epitopes for other viral proteins. In particular, we envision an integrated analysis of diverse betacoronaviruses, with the ultimate aim of producing a vaccine that guarantees broad protection against multiple members of this virus family.

## Methods

### Full-length molecular model of SARS-CoV-2 S glycoprotein

Our simulation system contained four membrane-embedded SARS-CoV-2 S proteins assembled from available resolved structures and models for the missing parts (S7 Fig). The spike head was modeled based on a recently determined structure (PDB ID: 6VSB [6]) with one RBD domain in an open conformation and glycans modeled according to [30]. The stalk connecting the S head to the membrane was modeled *de novo* as trimeric coiled coils, consistent with an experimental structure of the HR2 domain in SARS-CoV S (PDB ID: 2FXP [51]). The TMD as well as the cytosolic domain were modeled *de novo*. See S8 Fig for a view of the final model.

## Molecular dynamics simulations

We assembled four membrane-embedded full-length S proteins to form one large membrane patch. To maximize sampling while maintaining a biologically plausible S density [32, 33], we set the initial distance between centers of mass of the stalks of neighboring S to about 15 nm. This guaranteed at least 1 nm of spacing between any two of S's most extended glycans and thus no contacts between S in the initial configuration (cf. S5(A) Fig). Patches of comparably high density have been observed in experiments [23]. The full simulation system consisted of ∼4.1 million atoms. After 300 ns of equilibration, we performed production simulations of the four S proteins for 2.5 μs in the *NpT* ensemble with GROMACS 2019.6. We used the CHARMM36m protein and glycan force fields, in combination with the TIP3P water model, and sodium and chloride ions (150 mM). The time series of a series of parameters show that the system remains stable during the whole simulation (S2 Fig).

## Rigidity analysis

We quantified the local rigidity in terms of RMSF values. For each frame and each chain, the $C_\alpha$ atoms were rigid-body aligned to the average structure. For these aligned structures, the $C_\alpha$ RMSF was calculated. Then, for each residue of interest, we quantified the local flexibility as the average RMSF values of residues within 15 Å distance, weighted by the relative surface area of each residue [52]. These flexibility profiles were averaged over the four spike copies and three chains. The local rigidity was then defined as the reciprocal of the flexibility.

## Accessibility analysis

The accessibility of the S protein surfaces was probed by illuminating the protein in diffuse light, as detailed below, and by rigid-body docking of the Fab of the antibody CR3022 [10], as detailed in the S1 Text.

For the illumination analysis, rays of random orientation emanate from a half-sphere with radius 25 nm around the center of mass of the protein. They are absorbed by the first heavy atom they pass within 1.5 Å. Structures of single S collected at 10 ns intervals from the simulation of four S embedded in the membrane were each probed with $10^6$ rays. To quantify the effect of glycosylation, the analysis was performed with and without including the glycan shield. In addition, the effect of protein crowding on the ray accessibility was probed by considering all protein atoms of other S proteins with a minimum distance ≤3 nm from the illuminated S.

## Sequence variability analysis

To estimate the evolutionary variability of the S protein, we analyzed the aligned amino acid sequences released by the GISAID initiative on 25 May 2020 (https://www.gisaid.org/). We first built the consensus sequence with the most common amino acid (the mode) at each position across the whole data set. We then kept only 1273 amino acid long sequences, and filtered out corrupted sequences by discarding those having a Hamming distance from the consensus larger than 0.2. With the remaining 30,426 sequences, we estimated the conservation at each position [35]. Our conservation score is defined as the normalized difference between the maximum possible entropy and the entropy of the observed amino acid distribution at a given position, $\mathrm{cons}(i) = 1 + \sum_k p_k(i) \log p_k(i) / \log 20$, where $p_k(i)$ is the probability of observing amino acid $k$ at position $i$ in the sequence.

### Sequence-based epitope predictions

We estimated the epitope probability prediction by using the BepiPred 2.0 webserver (http://www.cbs.dtu.dk/services/BepiPred/), with an Epitope Threshold of 0.5 [36]. BepiPred 2.0 uses a random forest model trained on known epitope-antibody complexes.

### Consensus score for epitope prediction

We integrated the information of the different analyses into the consensus epitope score. We first applied a 3D Gaussian filter with $\sigma$ = 5 Å to the ray and docking scores. We then mapped each score to the interval [0, 1], with outliers mapped to the extremes listed in Table A in S1 Text. Finally, we multiplied the individual scores together to obtain the consensus score, which was also mapped to [0, 1].

## Supporting information

**S1 Text. Detailed Modelling Procedures, Detailed Methods, Consensus Score Parameters.** (PDF)

**S1 Fig. Spike domains and glycosylation.** (A) Domains of S. (B) Glycosylation pattern of S. Sequons are indicated with the respective glycans in a schematic representation for a fully glycosylated system ("full") and for resampled simulations containing only mannose-5 ("Man5"). (TIF)

**S2 Fig. Time series of various key parameters monitored during the simulation.** (A) Total potential energy, (B) Lennard-Jones energy, (C) Coulomb energy, (D-F) temperature, pressure, and volume of the simulation box. (G-J) Root-mean-square deviation (RMSD) over the course of the simulation, calculated for $C_\alpha$ carbons of the S body, CC1, HR2, and TMD, with respect to a reference configuration obtained after 300 ns of equilibration. Values for four spike proteins are shown with distinct colors. (TIF)

**S3 Fig. Impact of the glycosylation pattern on ray (A-C) and docking (D-G) accessibility.** (A-C) Number of ray hits without glycans ("no glycans"), with full glycans ("full", S1(B) Fig), and with full glycans and S protein crowding ("full CR"). (D-G) Monte Carlo rigid-body docking hits without glycans ("no glycans"), with Man5 glycans ("Man5", S1(B) Fig) and with full glycans ("full"), as well as with full glycans and S protein crowding ("full CR"). (TIF)

**S4 Fig. Comparison of the epitope candidates E3–E6 with previously characterized epitopes.** Glycans are shown in green licorice representation. Left panels: Epitope candidates shown in cartoon representation with purple color intensity indicating epitope consensus scores. Residues with epitope consensus score >0.1 are shown in licorice representation. Right panels: Epitopes described in previous works shown in cartoon and licorice representation, with higher purple color intensity indicating reported binding to multiple distinct antibodies. (TIF)

**S5 Fig. Effect of crowding on accessibility and epitope score.** (A) Ray, (B) docking and (C) consensus scores with (thick line) and without crowding being taken into account. (TIF)

**S6 Fig. Location and structural features of the epitope candidates E1–E9 on the S surface.** Epitope candidates are shown in red, orange and purple cartoon and licorice representation.

Neighboring residues are shown in grey cartoon representation.
(TIF)

**S7 Fig. Schematic illustration of the strategy used to obtain an atomistic model of the full-length S protein.** For clarity, we do not show the solvent and membrane.
(TIF)

**S8 Fig. Atomistic model of the full-length membrane-embedded S protein shown in cartoon representation.** The chains are differentiated by color. Palmitoylated cysteine residues are shown in pink licorice (only one chain shown for clarity). Glycans are shown in green licorice representation. We show a section of the membrane to highlight the transmembrane domain of S.
(TIF)

**S9 Fig. Spike-spike interactions and bending during MD simulation.** (A) Snapshots of the 4-spike system from above (top row) and in side-view (bottom row) at the beginning (left) and end (right) of the MD trajectory. While the transmembrane regions move relatively little, spike heads form spike-spike interactions because of significant bending at the "knee" (CC1—CC2 joint). These interactions persist on the simulation timescale. (B) Visualization of the glycans in the final configuration (blue sticks). Glycans mediate spike-spike contacts. (C and D) Maps of time-averaged spike-spike contact probability mediated by amino-acids (C) or amino-acids and glycans (D) from the MD trajectory (color bar: contact probability). Interactions are located exclusively on lateral faces of the spike head.
(TIF)

**S10 Fig. Consensus score analysis of "closed" spike.** (A, B) Accessibility, (C) rigidity and (D) consensus score calculated taking only into account the chains with down RBDs.
(TIF)

**S1 Movie. Atomistic molecular dynamics simulation trajectory of four S proteins embedded in a membrane.** The proteins and lipids are shown in surface representation. Glycans are represented by green van der Waals beads. Water and ions are omitted for clarity. 600 ns simulation time shown.
(MP4)

## Acknowledgments

We thank Martin Beck, Beata Turoňová, and Philipp S. Schmalhorst for stimulating discussions, the Max Planck Computing and Data Facility for providing computational resources, and the Leibniz Supercomputing Centre Munich for the SUPERspike computing allocation.

## Author Contributions

**Conceptualization:** Mateusz Sikora, Sören von Bülow, Florian E. C. Blanc, Michael Gecht, Roberto Covino, Gerhard Hummer.

**Data curation:** Mateusz Sikora, Sören von Bülow, Florian E. C. Blanc, Michael Gecht, Roberto Covino.

**Formal analysis:** Mateusz Sikora, Sören von Bülow, Florian E. C. Blanc, Michael Gecht, Roberto Covino, Gerhard Hummer.

**Funding acquisition:** Gerhard Hummer.

**Investigation:** Mateusz Sikora, Sören von Bülow, Florian E. C. Blanc, Michael Gecht, Roberto Covino.

**Methodology:** Mateusz Sikora, Sören von Bülow, Florian E. C. Blanc, Michael Gecht, Roberto Covino, Gerhard Hummer.

**Project administration:** Gerhard Hummer.

**Resources:** Roberto Covino, Gerhard Hummer.

**Software:** Mateusz Sikora, Sören von Bülow, Florian E. C. Blanc, Michael Gecht, Roberto Covino.

**Supervision:** Roberto Covino, Gerhard Hummer.

**Validation:** Mateusz Sikora, Sören von Bülow, Florian E. C. Blanc, Michael Gecht, Roberto Covino, Gerhard Hummer.

**Visualization:** Mateusz Sikora, Sören von Bülow, Florian E. C. Blanc, Michael Gecht, Roberto Covino.

**Writing – original draft:** Mateusz Sikora, Sören von Bülow, Florian E. C. Blanc, Michael Gecht, Roberto Covino, Gerhard Hummer.

**Writing – review & editing:** Mateusz Sikora, Sören von Bülow, Florian E. C. Blanc, Michael Gecht, Roberto Covino, Gerhard Hummer.

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
