## [Decision Letter · Decision Letter 0]

26 Nov 2020

Dear Dr. Hummer,

Thank you very much for submitting your manuscript "Computational epitope map of SARS-CoV-2 spike protein" for consideration at PLOS Computational Biology.

As with all papers reviewed by the journal, your manuscript was reviewed by members of the editorial board and by several independent reviewers. In light of the reviews (below this email), we would like to invite the resubmission of a significantly-revised version that takes into account the reviewers' comments.

We cannot make any decision about publication until we have seen the revised manuscript and your response to the reviewers' comments. Your revised manuscript is also likely to be sent to reviewers for further evaluation.

Sincerely,

Alexander MacKerell

Associate Editor

PLOS Computational Biology

Arne Elofsson

Deputy Editor

PLOS Computational Biology

Reviewer's Responses to Questions

**Comments to the Authors:**

Reviewer #1: In this manuscript the author present further analysis of MD simulations recently published in Turoñová et al, Science (2020), doi: 10.1126/science.abd5223. This additional information concerns:

1) the ability of the S glycans to shield the spike protein from the immune system and

2) the prediction of epitopes that can be targeted for vaccine design and development

I find that such information is indeed important to share publicly, however I am not entirely convinced that it justifies a second publication based on the same set of experiments.

Further to this, I found aspects on the aforementioned analysis that I found should be re-examined. More specifically, in regard to part 1) the analysis of the accessible surface was done through rigid-body docking the CR3022 antibody Fab region and also through a process called “ray” where the protein was illuminated by diffuse light to identify areas of higher accessibility. The results obtained through there two methods are dramatically different, for instance the glycan shield determines an accessibility reduction of 40% (ray) and 87% (docking). This is a very large difference as a reduction of 40% still signifies wide accessibility, meanwhile a reduction of 87% dramatically precludes it. I may have misunderstood this analysis, however the authors do not comment on such discrepancy.

In regards to part 2) the scoring function designed by the authors identifies a set of 9 epitopes that include 2 known ones. This point is highlighted as proof of the robustness of the score, yet those known epitopes are part of the glycan unshielded RBD and not so difficult to identify in general, as the RBD is the known target for the interaction with the ACE2 receptor and considering that the scoring function (rightfully) promotes unshielded regions. Notably, the score penalises these very regions in terms of flexibility, which is an aspect that was not addressed in the manuscript. A proof of the robustness of this epitope scoring function would be in my opinion to test some of the unknown predicted epitopes experimentally.

As a last point, I am afraid that the ‘trimming’ of the glycans to account for immature glycosylation is fundamentally wrong for two reasons. The first reason is that ER glycans would be large oligomannose types, such as Man9 and Man8, which are processed down to Man5 in the Golgi by alpha mannosidases; the Man5 conversion into complex N-glycans is then initiated by GnT1 also in the Golgi. Mammalian cells don’t have any paucimannose, which is more common in plants and insects. As for the second reason, the glycans 3D structure depends on their sequence and shorter versions do not have necessarily the same structure, similarly to how a protein region may not retain the same conformational propensity if trimmed down.

Reviewer #2: In this work, the authors present a massive, all-atom MD simulation of a patch of membrane of the SARS-CoV-2 virus with 4 spike proteins. Much of the analysis and so forth, as presented, is good. However I have some comments for the authors to consider.

My main comment is that the authors have a really unique opportunity to provide scientists with a view of the spike proteins in a very crowded environment. The spacing of the spike proteins – while this close range may exist in some rare instances – is quite different than the average structures, which place the spike proteins further apart.

That said, this crowded configuration does indeed exist and thus this work stands apart in its ability to inform others about what happens to the dynamics, the shielding, the flexibile hinges, etc. No other work does this.

But what is written somehow doesn’t capture the essence of this most intriguing aspect of the work. The authors only make a few very minor comments about this aspect. Yet, to me, as someone who also studies the spike protein – I think it is of utmost importance to analyze and I think it is quite interesting and worthy to be published.

In the introduction, the comment about the glycans playing a role and validated by experiment should be updated. The final published version of Casalino et al. provides experimental validation within the same paper.

I would have liked to see more discussion of the crowdedness of the system, and a finer analysis of what that means. How many contacts are made between the spikes here, and are all those contacts predominantly glycan mediated? The authors present a short analysis of this at a very high level, on page 4, but a map of the residues themselves (contact footprint, perhaps?) would be useful. Were these direct contacts made during system construction or do they form over time? In their now iconic image presented here as figure 1, e.g., the stalks are quite bent – I’ve always wondered – did the stalks start out that way, or did they move to that conformation over time, and are they sort of stuck like that, or is this terribly sampling limited? The authors don't provide any such information.

Otherwise I think the work is fine, and the epitope analysis, is interesting.

Overall I think this is an intriguing dataset, no doubt well-constructed and informative for folks, but some of what I thought was most interesting, was missing.

I also think the authors MUST deposit the full system models (PDB minimally, preferably actual simulation input files) as part of this work. There are likely other useful choices in terms of the many variable parameters, that the authors work will provide others, as we seek to understand things about this system.

Reviewer #3: This manuscript introduces a very interesting and a general approach to design novel antibodies to fight viral infections, including SARS-COV2. The approach is very elegant and based on combining multi-microsecond long molecular dynamics simulations of the target proteins in their native environment with bioinformatics-based analyses, which results in antibody-binding scores. Overall, the results in the manuscript are convincing and support the message of the authors. The manuscript is generally well-written and clear. It addresses a timely problem and should be of interest to PLOS Computational Biology readership and community in-general. Thus, I would be in favor of accepting the manuscript for publication in PLOS Computational Biology. However, I have some concerns that the author needs to address before it can be accepted.

Major Concerns:

1. Due to the plethora of on-going efforts to simulate full-length structure of SARS-COV2, including some of the recent work from these authors and Amaro lab, it is important to have a consensus on the full-length structure of SARS-COV2. Therefore, authors should discuss about their modeling approach in the light of recently published work by Amaro lab (ACS Central Science, 2020, 6, 10, 1722-1734).

2. Authors mentioned that during the MD simulations, S protein dynamically interacted with its neighboring copies. Does S protein forms stable interactions with the neighboring copies? Authors should provide a detailed analysis, maybe using clustering-based methods, to highlight the dynamic interactions between S proteins.

3. In order to perform bioinformatics-based epitope scoring, authors selected 220 x 4 snapshots from their 2.2μs-long MD simulations. How different were these snapshots? What was the criterion to select snapshots at 10ns intervals? This information will help readers not only to reproduce the data but also help them to intelligently apply this method to other important systems as well.

4. In-order to be exhaustive in their approach (which I really liked), authors performed epitope scoring under different glycan conditions. However, it seems like this analysis is based on the assumption that S protein dynamics will not be altered under different types of glycans. I think this is a strong assumption to make and authors should properly justify it.

5. Based on the starting S protein structure, 2 RBDs exists in down conformation and 1 RBD is in up conformation. Did authors captured conformation-dependent epitope scores for the regions residing on RBDs? Authors should include a discussion and epitope scoring data on this as well.

Minor Concerns:

1. Authors state that “On SARS-CoV-2 virions, S proteins occasionally form dense clusters, which may enhance the avidity of the interactions with human host cells [23]”. However, according to reference 23 (Figure 2A), there is no significant tendency to cluster. This point should be addressed.

2. Based on my understanding, epitope 5-6 seems to be in good correspondence with the recent pre-print ( https://www.biorxiv.org/content/10.1101/2020.08.08.238469v2 ) on nanobody design. Authors should include a discussion of their results with this CryoEM study. Interestingly, epitope 2 seems to be in good correspondence with the allosteric nanobody reported in the above pre-print.

3. In the methods section. authors state “the initial distance between the center of mass of the stalks if neighboring S was about 15nm.” This does not guarantee that ectodomain, to be specific RBDs are not interacting in the initial setup, thus making the system biased. Authors should indicate minimum distance between S proteins, in their initial setup.

4. Figure 4 is difficult to interpret and is too crowded, authors should make it more reader friendly.

5. Reference 30 in Supplementary Methods doesn’t seem to be on monte-carlo based rigid body docking. This should be corrected.

**Have all data underlying the figures and results presented in the manuscript been provided?**

Reviewer #1: Yes

Reviewer #2: **No: **authors should submit PDBs, PSF, input files for simulation

Reviewer #3: Yes

PLOS authors have the option to publish the peer review history of their article (what does this mean?). If published, this will include your full peer review and any attached files.

Reviewer #1: No

Reviewer #2: **Yes: **Rommie Amaro

Reviewer #3: No
---

## [Decision Letter · Decision Letter 1]

14 Feb 2021

Dear Dr. Hummer,

We are pleased to inform you that your manuscript 'Computational epitope map of SARS-CoV-2 spike protein' has been provisionally accepted for publication in PLOS Computational Biology.

Best regards,

Alexander MacKerell

Associate Editor

PLOS Computational Biology

Arne Elofsson

Deputy Editor

PLOS Computational Biology

Reviewer's Responses to Questions

**Comments to the Authors:**

Reviewer #1: I thank the authors for addressing my concerns.

Reviewer #2: The authors have done a very good job of responding to my critique and those of the other two reviewers. I think it can be published now!

Reviewer #3: The authors have addressed all of my concerns

**Have all data underlying the figures and results presented in the manuscript been provided?**

Reviewer #1: Yes

Reviewer #2: Yes

Reviewer #3: Yes

PLOS authors have the option to publish the peer review history of their article (what does this mean?). If published, this will include your full peer review and any attached files.

Reviewer #1: No

Reviewer #2: **Yes: **Rommie Amaro

Reviewer #3: **Yes: **Shashank Pant

---

## [Editor Report · Acceptance letter]

3 Mar 2021

PCOMPBIOL-D-20-01841R1 

Computational epitope map of SARS-CoV-2 spike protein

Dear Dr Hummer,

I am pleased to inform you that your manuscript has been formally accepted for publication in PLOS Computational Biology. Your manuscript is now with our production department and you will be notified of the publication date in due course.

With kind regards,

Andrea Szabo
